Deep evaluation of the evolutionary history of the Heat Shock Factor (HSF) gene family and its expansion pattern in seed plants

Liao Yiying 1
Liu Zhiming 2
Gichira Andrew W. 2
Yang Min 1
Mbichi Ruth Wambui 2
Meng Linping 1
Wan Tao 87418331@qq.com 2 3
1 Key Laboratory of Southern Subtropical Plant Diversity, Fairy Lake Botanical Garden, Shenzhen & Chinese Academy of Science , Shenzhen , China
2 Core Botanical Gardens/Wuhan Botanical Garden, Chinese Academy of Sciences , Wuhan , China
3 Sino-Africa Joint Research Centre, Chinese Academy of Sciences , Wuhan , China
Basile Adriana
Electronic publication date: 2022 Aug 9
Publication date: 2022
Volume: 10
Electronic Location ID: e13603
Received 2021 Jan 7; Accepted 2022 May 26
Copyright: ©2022 Liao et al.
Copyright year: 2022
Copyright holder: Liao et al.
License: This is an open access article distributed under the terms of the Creative Commons Attribution License, which permits unrestricted use, distribution, reproduction and adaptation in any medium and for any purpose provided that it is properly attributed. For attribution, the original author(s), title, publication source (PeerJ) and either DOI or URL of the article must be cited.
License URL: https://creativecommons.org/licenses/by/4.0/

Keywords: Heat shock factor, Gene family evolution, Diversification, Lineage-specific expansions, Whole genome duplication

Funding: National Natural Scientific Foundation of China 31870206 31670369 Innovation of Science and Technology Commission of Shenzhen Foundation JCYJ201206151530054 The Scientific Research Foundation of Fairy Lake Botanical Garden Scientific Research Program of Sino-Africa Joint Research Center SAJL201607 This work was supported by National Natural Scientific Foundation of China (Grant No. 31870206, 31670369), the Innovation of Science and Technology Commission of Shenzhen Foundation (Grant No. JCYJ201206151530054), the Scientific Research Foundation of Fairy Lake Botanical Garden and the Scientific Research Program of Sino-Africa Joint Research Center (Grant No. SAJL201607). The funders had no role in study design, data collection and analysis, decision to publish, or preparation of the manuscript.

==============================
Heat shock factor (HSF) genes are essential in some of the basic developmental pathways in plants. Despite extensive studies on the structure, functional diversification, and evolution of HSF genes, their divergence history and gene duplication pattern remain unknown. To further illustrate the probable divergence patterns in these subfamilies, we analyzed the evolutionary history of HSF genes using phylogenetic reconstruction and genomic syntenic analyses, taking advantage of the increased sampling of genomic data from pteridophytes, gymnosperms and basal angiosperms. We identified a novel clade that includes HSFA2, HSFA6, HSFA7, and HSFA9 with a complex relationship, which is very likely due to orthologous or paralogous genes retained after frequent gene duplication events. We hypothesized that HSFA9 derives from HSFA2 through gene duplication in eudicots at the ancestral state, and then expanded in a lineage-specific way. Our findings indicate that HSFB3 and HSFB5 emerged before the divergence of ancestral angiosperms, but were lost in the most recent common ancestors of monocots. We also presumed that HSFC2 derives from HSFC1 in ancestral monocots. This work proposes that during the radiation of flowering plants, an era during which there was a differentiation of angiosperms, the size of the HSF gene family was also being adjusted with considerable sub- or neo-functionalization. The independent evolution of HSFs in eudicots and monocots, including lineage-specific gene duplication, gave rise to a new gene in ancestral eudicots and monocots, and lineage-specific gene loss in ancestral monocots. Our analyses provide essential insights for studying the evolutionary history of this multigene family.

Introduction

Heat shock factors (HSFs) play an important role in improving the thermotolerance of plants. They function as the central regulators of heat shock protein expression and other heat shock-induced gene expression. HSFs are the direct transcriptional activators of genes regulated by thermal stress. They encode heat shock proteins to protect cells against proteotoxic damage under heat stress (HS; Hu, Hu & Han, 2009; Ahuja et al., 2010; Ohama et al., 2017). HSFs have been identified in most eukaryotes and non-plant organisms. HSFs also participate in the growth and development of cells (Åkerfelt, Morimoto & Sistonen, 2010; Scharf et al., 2012). HSFs have been widely studied in plants, especially angiosperms, and have been found to be critical under various environmental stressors (Scharf et al., 2012). The number of HSF genes varies widely among plants with green algae having just one or two HSF genes and angiosperms having more than 50 (Wang et al., 2018). HSFs generally contain the DNA binding domain (DBD), the oligomerization domain (OD), the nuclear localization signal (NES), and the C-transcriptional activation domain (CTD) (Scharf et al., 2012; Guo et al., 2016). Based on the topology of these domains, HSFs are normally classified into three groups: HSFA, HSFB and HSFC. These three groups of HSFs are further divided into 16 subfamilies which are distinguished in angiosperms, including the HSFA group (A1-A9), HSFB group (B1-B5) and HSFC group (C1-C2) (Nover et al., 2001; Hu, Hu & Han, 2009; Scharf et al., 2012; Qiao et al., 2015). HSFC was identified for the first time in the first overview of HSFs, presented in Arabidopsis thaliana by Nover et al. (2001). Many valuable summaries followed, including compiled data from nine angiosperm species, and over 50 plant species showing the structure, function and evolution of HSFs (Scharf et al., 2012; Wang et al., 2018). These reports point out that HSF family members and their functions are greatly diverged among higher plant lineages in response to environmental stressors. However, the evolutionary relationships among these subfamilies are still unknown; some of the deepest nodes of the HSF phylogeny tree, such as the positions of HSFB5 and HSFA9, also remain unclear. In previous studies, HSFB5 was either placed with HSFA5 or other HSFB members, and HSFA9 may be clustered with HSFA2 or HSFA7 (Scharf et al., 2012; Wang et al., 2018). This is likely due to the limited HSF data available in representative seed plant lineages including gymnosperms and basal angiosperms. It is also partially attributed to the unpredictable gene copy turnover after recurring gene duplication events at tandem or genome-wide level.

In this study, we expanded the data collection to basal angiosperms, gymnosperms, and pteridophytes to reconstruct the diversification history of HSFs during seed plant evolution. We also detected the syntenic relationships of HSFs across a wide range of species, thus providing crucial information to address fundamental questions on their evolutionary history. We then estimated the divergence time of the derived genes from their ancestors based on a reliable gene orthology. Our results present critical evidence that help to explain the expansion of HSF subfamilies in seed plant lineages.

Methods

Identification of HSFs and phylogenetic analysis

Here, we sampled 23 species representing three main taxa for pteridophytes, gymnosperms and basal angiosperms including seven genomes and 17 transcriptomes, (Table S1). Most of the transcriptome data was obtained from the National Center for Biotechnology Information (NCBI, https://www.ncbi.nlm.nih.gov) (Data from Ran et al., 2018). Multiple databases were screened for the genome assemblies including: ConGenIE (http://congenie.org/), GigaDB (http://gigadb.org/dataset/100209), Dryad (https://datadryad.org/stash/dataset/), WaterlilyPond (http://eplant.njau.edu.cn/waterlily/), FernBase (https://www.fernbase.org/), and the Liriodendron chinense database (http://120.78.193.56:8000/). To increase the reliability of the data, we analyzed both the genomes and transcriptomes of Ginkgo biloba in this study. The methods of our RNA-seq dataset analysis were drawn from a study by Ran et al. (2018). We used the predicted proteome of each genome as a query to search for HSF-type DBD domains (HSF_DNA_bind_PF00447) from Pfam-A.hmm (Pfam release 32.0) using PfamScan software (https://www.ebi.ac.uk/Tools/pfa/pfamscan/), which were considered as candidate genes. We then extracted the amino acid sequences of the HSFs. We also downloaded 537 HSF sequences extracted from 23 plant species (Table S2) representing seven main taxa in the Heatster database (http://www.cibiv.at/services/hsf) and used them in BLAST searches for analyzed species to further identify candidate HSF proteins. For those candidate sequences, we examined the facticity of DBDs and ODs using the SMART 7 software (Letunic, Doerks & Bork, 2012) (http://smart.embl-heidelberg.de/) and the HEATSTER website (https://applbio.biologie.uni-frankfurt.de/hsf/heatster/). The candidate proteins without an integrated DBD domain or HR-A/B domain were removed.

For the phylogenetic reconstruction in this study, we used MUSCLE (http://www.drive5.com/muscle) to conduct the alignment of the candidate genes. Phylogenetic trees were generated using both the NJ and the ML methods. The NJ tree was constructed by TreeBeST (version 1.9.2, http://treesoft.sourceforge.net/treebest.shtml, parameters: -t mm –b 100). Approximately maximum-likelihood (ML) phylogenetic trees were constructed using FastTree (version 2.1.11, http://www.microbesonline.org/fasttree/treecmp.html, with default parameters). Then, phylogenetic analyses were conducted using RaxML version 8.0.19 (Stamatakis, 2014) with 100 bootstraps, the PROTGAMMAAUTO model, and maximum likelihood reconstruction using rapid hill-climbing and rapid bootstrap analyses (-f ad). Phylogenetic trees were examined and manipulated with Evolview v2 (He et al., 2016). We classified the HSF subfamilies using both the phylogenetic tree and the annotation from the HEATSTER website. Some results from the HEATSTER website were inconsistent with the phylogenetic tree, so we performed follow-up checks to confirm the subfamily classification. The final results were based on the appearance of domain characteristic motifs.

We used all HSFA, HSFB, and HFSC genes identified for phylogenetic tree reconstruction in order to better understand the evolutionary relationship within subfamilies and for in depth phylogenetic analyses of the HSFB clade and HSFA-HSFC clade. To understand the complicated evolutionary relationship of the HSFA2, HSFA6, HSFA7, and HSFA9 clades of subfamilies and the HSFC clade, we extracted those two group genes for phylogenetic tree reconstruction, with Chlamydomonas reinhardtii used as an outgroup. Our methods for protein sequence alignments and phylogenetic analyses followed the same steps as previously outlined.

Synteny analysis and molecular dating analyses

We used MCScanX (Wang et al., 2012) to detect the gene replication events and included a total of 21 plant genomes in a synteny analysis covering green algae, mosses, ferns, gymnosperms, basal angiosperms and angiosperms (Table S3). We analyzed all protein models from these genomes for all possible intra- and inter-species genome-wide comparisons and downloaded all genome annotation and corresponding protein sequences for those species. Homologous genes are classified as either orthologous in different species if they are separated by a speciation event, or paralogous in the same species if they are separated by a gene duplication event. We identified the paralogous and orthologous genes in or between those genomes through synteny detection using MCScanX with default parameters (minimum match size for a collinear block = five genes, max gaps allowed = 25 genes). The output files from all the intra- and inter-species comparisons were integrated into a single file named “Total_Synteny_Blocks”, including the headers “Block_Index”, “Locus_1”, “Locus_2”, and “Block_Score”, which served as the database file. We performed the all-against-all protein sequence comparisons necessary for MCScanX using DIAMOND v 0.8.25 (Buchfink, Xie & Huson, 2015). The gene list containing all candidate HSF genes was queried against the “Total_Synteny_Blocks” file. We used these results to identify whether or not HSF genes exist in a syntenic block. We chose eight representative species for gymnosperms (Gnetum montanum, Ginkgo biloba), basal angiosperms (L. chinense, Amborella trichopoda), monocots (Oryza sativa, Zea mays), and eudicots (A. thaliana, Solanum lycopersicum) to do a synteny analysis between species on close taxa. The methods and procedures used were the same as those previously outlined.

The HSFC1-C2 subfamily genes and the HSFA2 and HSFA9 subfamily genes were extracted from the database and used to estimate divergence time. We calibrated a relaxed molecular clock on the node and found the divergence time of monocots and eudicots to be between 140 Mya (a minimum age) and 200 Mya (a maximum age) (represented by the divergence of A. thaliana and O. sativa, Gensel & Andrews, 1984). We performed a Bayesian dating analysis in the Markov chain Monte Carlo (MCMC) tree (Yang, 2007) using an approximate likelihood calculation for the branch lengths, an auto-correlated model of among-lineage rate variation, the GTR substitution model, and a uniform prior on the relative node times. We used Markov chain Monte Carlo sampling to estimate posterior distributions of node ages, with samples drawn every two steps over 200,000 steps following a burn-in of 10,000 steps. We could then trace the gene duplication time based on the resulting gene divergence times.

Table 1 The species used for phylogenetic tree construction, and the category of HSFs.

			Category of HSFs	
Taxonomy	Species	Abbreviation	A1	A2	A3	A4	A5	A6	A7	A8	A9	B1	B2	B3	B4	B5	C1	C2	HSF like (N.C.)	Total	
Chlorophyta	Chlamydomonas reinhardtii	Chlre	1	1	0	0	0	0	0	0	0	0	0	0	0	0	0	0	2	4	
Chlorophyta	Volvox carteri	Volca	0	1	1	0	0	0	0	0	0	0	0	0	0	0	0	0	0	2	
Bryophyta	Physcomitrella patens	Phypa	4	0	0	0	0	0	0	0	0	4	0	0	0	0	0	0	0	8	
Pteridophyta	Selaginella moellendorffii	Selmo	4	0	0	0	0	0	0	0	0	2	0	0	1	0	0	0	0	7	
Pteridophyta	Azolla filiculoides	Azofi	7	0	0	0	0	0	0	0	0	3	0	0	3	0	0	0	1	14	
Pteridophyta	Ceratopteris gametophytes a	Cerga	3	1	0	0	0	0	0	0	0	2	0	0	0	0	0	0	3	9	
Pteridophyta	Lygodium japonicum a	Lygja	5	0	0	0	0	0	0	0	0	2	0	0	0	0	0	0	6	13	
Pteridophyta	Pteridium aquilinum a	Pteaq	1	1	0	0	0	0	0	0	0	3	0	0	0	0	0	0	5	10	
Pteridophyta	Salvinia cucullata	Salcu	7	1	0	0	0	0	0	0	0	4	0	0	3	0	0	0	0	15	
Gymnosperm	Abies firma a	Abifi	2	1	0	0	0	0	0	0	0	2	1	0	2	0	0	0	3	11	
Gymnosperm	Araucaria cunninghamii a	Aracu	4	0	0	0	0	0	0	0	0	4	1	0	1	0	0	0	3	13	
Gymnosperm	Cephalotaxus sinensis a	Cepsi	3	0	0	0	0	0	0	0	0	2	1	0	1	0	0	0	1	8	
Gymnosperm	Cycas revoluta a	Cycre	4	0	0	0	0	0	0	0	0	1	1	0	1	0	0	0	1	8	
Gymnosperm	Ephedra equisetina a	Epheq	4	0	0	0	0	0	0	0	0	0	0	0	1	0	0	0	2	7	
Gymnosperm	Ginkgo biloba	Ginbi	5	0	0	0	0	0	0	0	0	2	0	0	1	0	0	0	0	8	
Gymnosperm	Ginkgo biloba a	GinbiR	4	0	0	0	0	0	0	0	0	0	1	0	2	0	0	0	2	9	
Gymnosperm	Gnetum montanum	Gnemo	6	0	0	0	0	0	0	0	0	1	0	0	4	0	0	0	13	24	
Gymnosperm	Metasequoia glyptostroboides a	Metgl	6	0	0	0	0	0	0	0	0	2	1	0	1	0	0	0	2	12	
Gymnosperm	Picea abies	Picab	4	0	0	0	0	0	0	0	0	7	1	0	2	0	0	0	5	19	
Gymnosperm	Picea abies a	PicabR	3	1	0	0	0	0	0	0	0	2	1	0	0	0	0	0	2	9	
Gymnosperm	Picea glauca	Picgl	4	0	0	0	0	0	0	0	0	11	1	0	2	0	0	0	0	18	
Gymnosperm	Pinus taeda	Pinta	7	1	0	0	0	0	0	0	0	15	1	0	4	0	0	0	20	48	
Gymnosperm	Pinus taeda a	PintaR	4	0	0	0	0	0	0	0	0	2	1	0	2	0	0	0	1	10	
Gymnosperm	Podocarpus macrophyllus a	Podma	3	0	0	0	0	0	0	0	0	3	1	0	0	0	0	0	3	10	
Gymnosperm	Sciadopitys verticillata a	Scive	2	0	0	0	0	0	0	0	0	3	1	0	1	0	0	0	3	10	
Gymnosperm	Taxus chinensis a	Taxch	5	0	0	0	0	0	0	0	0	2	1	0	0	0	0	0	1	9	
Gymnosperm	Welwitschia mirabilis a	Welmi	8	0	0	0	0	0	0	0	0	1	0	0	1	0	0	0	2	12	
Gymnosperm	Zamia furfuracea a	Zamfu	4	0	0	0	0	0	0	0	0	2	1	0	1	0	0	0	1	9	
Basal angiosperms	Amborella trichopoda	Ambtr	2	1	1	0	1	1	0	0	0	1	1	0	2	1	0	0	2	13	
Basal angiosperms	Liriodendron chinense	Lirch	2	2	1	1	2	1	1	0	0	2	2	1	1	1	1	0	0	18	
Basal angiosperms	Nymphaea colorata	Nymco	3	1	1	1	1	0	1	0	0	1	2	0	3	1	2	0	4	21	
Eudicots	Arabidopsis thaliana	Arath	4	1	1	2	1	2	2	1	0	1	2	1	1	0	1	0	4	24	
Eudicots	Cajanus cajan	Cajca	2	1	1	2	1	2	1	1	1	2	2	1	4	1	1	0	4	27	
Eudicots	Citrullus lanatus	Citla	1	2	1	3	1	2	0	2	1	1	2	2	3	1	2	0	0	24	
Eudicots	Mimulus guttatus	Mimgu	2	1	1	2	0	2	0	1	0	0	2	1	2	1	1	0	5	21	
Eudicots	Nelumbo nucifera	Nelnu	4	2	1	2	1	1	0	1	0	2	2	2	3	2	2	0	0	25	
Eudicots	Populus trichocarpa	Poptr	3	1	1	3	2	4	0	2	1	1	3	2	4	2	1	0	4	34	
Eudicots	Prunus persica	Prupe	2	1	1	2	0	2	0	1	1	1	2	1	1	1	1	0	3	20	
Eudicots	Solanum lycopersicum	Solly	4	1	1	3	1	2	0	1	3	1	2	2	2	1	1	0	1	26	
Monocots	Brachypodium distachyon	Bradi	1	3	1	2	1	2	2	1	0	1	3	0	3	0	2	2	2	26	
Monocots	Oryza brachyantha	Orybr	0	3	0	2	1	2	2	1	0	1	1	0	3	0	2	1	3	22	
Monocots	Oryza sativa	Orysa	1	4	1	2	1	2	2	1	0	1	3	0	4	0	2	2	3	29	
Monocots	Phoenix dactylifera	Phoda	7	3	2	2	2	2	0	0	0	2	4	0	3	0	1	2	1	31	
Monocots	Phyllostachys heterocycla	Phyhe	1	4	2	3	2	3	1	0	0	2	2	0	4	0	2	1	14	41	
Monocots	Sorghum bicolor	Sorbi	1	3	1	1	1	2	2	1	0	1	3	0	3	0	2	2	3	26	
Monocots	Triticum urartu	Triur	1	5	1	1	1	0	0	1	0	1	0	0	0	0	2	1	6	20	
Monocots	Zea mays	Zeama	2	2	1	3	1	2	2	2	0	2	4	0	2	0	2	2	13	40	
Notes.

a The data from transcriptomes. N.C. the sequence only contains some of the necessary domains for a heat shock transcription factor and therefore it could not be classified.

Results

The phylogeny and evolution of HSFs in land plants

A total of 670 HSF sequences from 44 species were used for the phylogenetic analysis (Tables S1 and S2). We identified 287 new candidate HSF sequences from 24 species, with 228 of those divided into known subfamilies (A1-A9, B1-B5, C1-C2) (Table 1) on the HEATSTER website. Across the comprehensive samples studied, the number of HSF gene subfamilies identified varied greatly, ranging from two in chlorophyta to 30 in angiosperms. The unrooted phylogenetic tree inferred from amino acid sequences was well resovled to three main clades: HSFA, HSFB and HSFC (Fig. 1, Figs. S1–S4). The newly identified HSF genes were re-confirmed on a phylogenic tree. Most subfamilies of clades (A3, A4, A5, A8, A9, B2, B3, B5, C1, C2) were accordingly recovered, while the relationships between these clades were weakly supported. The HSF subfamilies displayed a strong diversification in structure, composition and function (Scharf et al., 2012; Guo et al., 2016; Wang et al., 2018), thus, significant genetic differentiation between clades, especially for HSFA and HSFB were likely resulted from the unstable topology observed. The HSFA group was found in all sampled taxa, while the HSFB group was absent in chlorophyta, and the HSFC group was only present in the angiosperms.

The HSFA group contains major regulators in the HS response of plants (Wang et al., 2018), and, as a result of diversification during plant evolution, displayed variations in different taxa. Interestingly, the A4-A9 subfamily clades are only occurred in angiosperms and A9 genes are only identified in Eudicots. Some subfamilies clustered as a branch, such as A3, A4, A5, and A9, while others were clustered as several branches (Figs. S1 and S5). HSFA1 is a master regulator which cannot be replaced by any other HSF (Scharf et al., 2012) and probably be the most ancient HSFA group. Although all the HSFA1 genes with HSFA8 in angiosperms clustered as a clade, most of the HSFA1 genes from pteridophytes and gymnosperms were dispersed into several clades. The deep divergence of HSFA1 in pteridophytes and gymnosperms indicates the early diversification of HSFA1 before the radiation of all seed plants. Meanwhile, HSFA2, HSFA6, HSFA7, and HSFA9 were blended into a complex clade, and HSFA9 formed a monophyletic group, but others remain unclear. We also noticed that the HSFA2 gene and the HSFA6 gene clustered together with very little genetic difference in some angiosperm species such as O. sativa, Phoenix dactylifera, Citrullus lanatus, as the HSFA6 and HSFA7 did in C. lanatus. The relationship between the HSFA4 clade and the HSFA5 clade was closer in the tree, with two HSFA5 genes sneaked into the HSFA4 clade. It has previously been suggested a close relationship between HSFA3 and HSFC, however, due to the increased number of ferns and gymnosperms, one HSFA1 clade of gymnosperms, rather than HSFA3, was clustered with HSFC.

Figure 1 An unrooted Maximum-Likelihood tree showing the hylogeny and classification of 670 HSFs sequences from 44 species representing seven main taxa including chlorophyta, bryophyta, peridophyta, gymnospermae, basal angiosperms, eudicots and monocots.

The information of species and sequences accession numbers used for the tree are listed in File S1. HSFA, HSFB and HSFC are clustered into three main clades. The clade of subfamilies HSFA2-7, HSFA8 and HSFA9, HSFB2-5, and HSFC1 and HSFC2, were shown over relevant branches with different colors. The three groups HSFA, HSFB, and HSFC were highlighted with shades of different colors. The scale bar represents amino acid substitutions per site.

HSFC only displayed the pattern as the angiosperms clade clustered to one clade of HSFA1 in gymnosperms (Figs. S1, S2, S3 and S5). It is assumed that a duplication event occurred in the ancestral angiosperms which could have contributed to the rise of HSFC. HSFC1 is a common gene subfamily and varies in gene numbers between monocots and eudicots (Table 1, Fig. S5); there are usually two members in most monocots and only one member in eudicots. These results indicate that HSFC experienced steady expansion during the evolution of monocots, and may be involved in important developmental pathways (Wang et al., 2018). Notably, HSFC2 was only present in monocots, but HSFC1 was present in all angiosperm species except for A. trichopoda. In monocots, HSFC1 and HSFC2 clustered together with strong support. HSFC1-HSFC2 clade of monocots group to HSFC1 of eudicots, based with HSFC1 of basal angiosperms. This suggests that HSFC2 is the result of recent duplication that occurred early in the divergence between monocots and eudicots.

Contrary to a previous study (Wang et al., 2018), the results of this study suggest that the HSFB subfamily (HSFB1-HSFB5) is moderately supported as a monophyletic group (Figs. S1 and S6). HSFB1, HSFB2, and HSFB4 have been widely observed across land plants, while both HSFB3 and HSFB5 are only present in the eudicots and basal angiosperms. Although HSFB5, unlike other subfamily members of the HSFB group, has a conserved tetrapeptide LFGV in the C-terminal domain, it is closely related to HSFB3 (Fig. 1). Additionally, the number of HSFB1 genes in gymnosperms is far more than that in angiosperms, with the common number reduced from 3 or 4 in gymnosperms to 1 or 2 in angiosperms (Table 1). In particular, the number of HSFB1 genes in conifers (Picea abies, Pinus taeda, Picea glauca) is significantly increased than that of other seed plants. Multiple copies of the HSFB1 gene in P. abies, P. taeda, and P. glauca clustered and formed a strongly supported monophyletic group. This result indicates that the evolution of these three conifers probably involved both polyploidy and repetitive element activity (Drewry, 1988; Ahuja, 2005; Li et al., 2015). The multi-copy genes may be attributed to two whole genome duplication (WGD) events in the ancestry of major conifer clades (Li et al., 2015). Though many angiosperm lineages have experienced additional rounds of genome duplication (Soltis, Visger & Soltis, 2014; Jiao et al., 2014; Li et al., 2015), there is no obvious proliferation in the member. This result is consistent with the speculation that WGD in angiosperms did not give rise to a remarkable expansion of HSFB1 genes. The HSFB1 genes in angiosperms, gymnosperms and pteridophytes were independently found in different branches, which suggests that HSFB1 is an ancient group which diverged during the evolutionary history of different taxa. HSFB1 genes in gymnosperms experienced several expansions including ancient duplication, while HSFB1 genes in angiosperms rarely retained duplication except for a few recent duplicates. All HSFB2 genes in gymnosperms and angiosperms clustered as a group, respectively. We were unable to trace out a remarkable expansion in gymnosperms, but more than two genes in angiosperms were assumed to be the result of recent duplication. In some species, such as Selaginella moellendorffii, we observed that some genes identified as different subfamilies, such as HSFB1 and HSFB4, and have genetic similarities to highly supported clades. The complicated relationship of these two subfamilies may be a result of recent duplication events. In this study, the HSFB3 and HSFB5 subfamilies were only present in eudicots and basal angiosperms. This is likely the result of duplication events occurring in ancestral angiosperms with subsequent loss of paralogue genes in the monocots.

Gene duplication analysis

To examine the expansion patterns and genetic divergences of the HSF family, a synteny analysis was performed to identify gene duplication events across 21 species (Table S3). We also conducted a synteny analysis between different species on the closely related taxa.

Gene duplication events were identified in 11 species including pteridophytes, basal angiosperms, monocots and eudicots (Table 2). In green algae, moss, and gymnosperms, we did not detect any HSF genes in synteny blocks. In S. moellendorffii, the only non-seed plant analyzed, we identified one pair of duplication genes. These two genes, ‘SelmoHSFB1b’ and ‘SelmoHSFB4,’ belong to different subclasses of the HSF gene family which were observed as being syntenic to each other. We speculate that these genes may be derived from a duplication event and have evolved with differences at the gene sequence level. In L. chinense, the only basal angiosperm analyzed, we identified five pairs of duplication genes with four of those five pairs from the same gene subclass (HSFA2, HSFB1, HSFB2, HSFC1) and the remaining pair from a different gene subclass (HSFA4-HSFA5). Gene duplication events were detected in all sampled eudicot and monocot species. In five eudicots (A. thaliana, Populus trichocarpa, Prunus persica, S. lycopersicum, Mimulus guttatus), we identified 29 pairs of duplication genes out of which 33 pairs belonged to the same gene subclasses (HSFA1, HSFA4, HSFA5, HSFA6, HSFA8, HSFB2, HSFB3, HSFB4, HSFB5) and four pairs belonged to different gene subclasses (HSFA2-HSFA9, HSFA6-HSFA7). In four monocots (O. sativa, Sorghum bicolor, Z. mays, Brachypodium distachyon), we also identified 29 pairs of duplication genes out of which 33 pairs belonged to the same gene subclasses (HSFA1, HSFA2, HSFA4, HSFA6, HSFB1, HSFB2, HSFB4, HSFC1, HSFC2) and four pairs belonged to different gene subclasses (HSFA2-HSFA6, HSFB1-HSFB2, HSFB2-HSFB4). In general, all HSF gene subfamilies except HSFA3 showed the signature of gene duplication. These results also demonstrated that gene pairs from different subfamilies, such as HSFA2-HSFA6, HSFA2-HSFA9, HSFA4-HSFA5, HSFA6-HSFA7, HSFB1-HSFB4, HSFB1-HSFB2, and HSFB2-HSFB4, were paralogous gene pairs.

Table 2 The detected paralogous genes within different species.

Order	Species	Paralogous genes Types	
Pteridophyta	Selaginella moellendorffii	HSFB1-HSFB4	
Basal angiosperms	Liriodendron chinense	HSFC1-HSFC1, HSFA2-HSFA2, HSFA4-HSFA5, HSFB1-HSFB1, HSFB2-HSFB2	
Eudicots	Arabidopsis thaliana	HSFA1-HSFA1, HSFA4-HSFA4, HSFA6-HSFA6, HSFA6-HSFA7	
Populus trichocarpa	HSFA1-HSFA1, HSFA4-HSFA4, HSFA5-HSFA5, HSFA6-HSFA6, HSFA8-HSFA8, HSFA9-HSFA2, HSFB2-HSFB2, HSFB3-HSFB3, HSFB4-HSFB4, HSFB5-HSFB5	
Prunus persica	HSFA2-HSFA9, HSFA6-HSFA6, HSFB2-HSFB2	
Solanum lycopersicum	HSFA1-HSFA1, HSFA4-HSFA4, HSFA6-HSFA6, HSFA9-HSFA2, HSFB2-HSFB2, HSFB3-HSFB3	
Mimulus guttatus	HSFB4-HSFB4	
Monocots	Oryza sativa	HSFA2-HSFA2, HSFA6-HSFA2, HSFB2-HSFB2, HSFB4-HSFB4, HSFC2-HSFC2	
Sorghum bicolor	HSFA2-HSFA2, HSFA2-HSFA6, HSFA6-HSFA6, HSFB2-HSFB2, HSFC2-HSFC2	
Zea mays	HSFA1-HSFA1, HSFA2-HSFA2, HSFA4-HSFA4, HSFB1-HSFB1, HSFB2-HSFB1, HSFB2-HSFB2,
HSFB2-HSFB4, HSFC1-HSFC1, HSFC2-HSFC2	
Brachypodium distachyon	HSFA2-HSFA2, HSFA6-HSFA6, HSFB2-HSFB2, HSFB4-HSFB4, HSFC2-HSFC2	

Beyond that, synteny analysis among different species identified the orthologous genes in different taxa (Table 3). In detail, only HSFA1 genes from different sources were found as orhologous genes between two gymonosperms (G. montanum, G.biloba). As a result of the analysis of gymnosperms (G. biloba) and basal angiosperms (L. chinense), HSFA1, HSFA4, and HSFA5 were detected as orthologous genes. Though the analysis among basal angiosperms (A. trichopoda, L. chinense) and eudicots (S. lycopersicum, A. thaliana) found several orthologous genes, such as HSFA6-HSFA7, HSFA4-HSFA5, HSFA2-HSFA9, and HSFB2-HSFB5, among basal angiosperms and monocots (O. sativa, Z. mays), we only identified out HSFA2-HSFA6 and HSFA2-HSFA7 as orthologous genes. Interestingly, the analysis of eudicots-monocots reveal a consistent pattern as basal angiosperms-monocots with HSFA1-HSFA5 and HSFA2-HSFA7 being identified as orthologous genes in monocots and HSFA6-HSFA7 as orthologous genes in eudicots.

Table 3 The ortologous gene clusters detected between different species.

Pairwise_Taxa	Pairwise_Species	Ortologous gene Types	
Gymnosperm-Gymnosperm	Gnetum montanum-Ginkgo biloba	HSFA1-HSFA1	
Gymnosperm-Basal angiosperms	Amborella trichopoda-Ginkgo biloba	HSFA1-HSFA1	
Liriodendron chinense-Ginkgo biloba	HSFA4-HSFA1, HSFA5-HSFA1	
Basal angiosperms-Basal angiosperms	Liriodendron chinense-Amborella trichopoda	HSFA1-HSFA1,HSFA2-HSFA2,HSFA3-HSFA3,HSFA5-HSFA5,HSFA6-HSFA6,HSFB2-HSFB2,HSFB5-HSFB5	
Basal angiosperms-Eudicots	Arabidopsis thaliana-Amborella trichopoda	HSFA1-HSFA1, HSFA5-HSFA5, HSFA6-HSFA6, HSFA6-HSFA7, HSFB2-HSFB2,	
Arabidopsis thaliana-Liriodendron chinense	HSFA1-HSFA1, HSFA2-HSFA2, HSFA4-HSFA4, HSFA4-HSFA5, HSFB1-HSFB1, HSFB2-HSFB2, HSFB3-HSFB3, HSFC1-HSFC1, HSFC1-HSFC1	
Liriodendron chinense-Solanum lycopersicum	HSFA1-HSFA1, HSFA2-HSFA2, HSFA2-HSFA9, HSFA4-HSFA4, HSFA4-HSFA5, HSFB1-HSFB1, HSFB2-HSFB2, HSFB3-HSFB3, HSFB4-HSFB4, HSFC1-HSFC1, HSFC1-HSFC1	
Amborella trichopoda-Solanum lycopersicum	HSFA1-HSFA1, HSFA2-HSFA2, HSFA2-HSFA9, HSFA5-HSFA5, HSFA6-HSFA6, HSFB5-HSFB2, HSFB5-HSFB5,	
Basal angiosperms-Monocots	Zea mays-Amborella trichopoda	HSFA2-HSFA6, HSFA3-HSFA3, HSFA6-HSFA6, HSFB2-HSFB2	
Zea mays-Liriodendron chinense	HSFB1-HSFB1,HSFB2-HSFB2,HSFB4-HSFB4	
Oryza sativa-Amborella trichopoda	HSFA2-HSFA7, HSFA3-HSFA3, HSFA6-HSFA2, HSFA6-HSFA6, HSFB2-HSFB2	
Oryza sativa-Liriodendron chinense	HSFA4-HSFA4, HSFA7-HSFA2, HSFB1-HSFB1, HSFB2-HSFB2, HSFB4-HSFB4	
Eudicots-Monocots	Oryza sativa-Arabidopsis thaliana	HSFA6-HSFA2, HSFA6-HSFA6, HSFA7-HSFA2	
Oryza sativa-Solanum lycopersicum	HSFA2-HSFA6, HSFA4-HSFA4, HSFA6-HSFA6, HSFA7-HSFA2, HSFB1-HSFB1, HSFB2-HSFB2, HSFB4-HSFB4,	
Zea mays-Arabidopsis thaliana	HSFA2-HSFA6	
Zea mays-Solanum lycopersicum	HSFA2-HSFA6, HSFA6-HSFA6, HSFB1-HSFB1, HSFB2-HSFB2,	
Monocots-Monocots	Oryza sativa-Zea mays	HSFA1-HSFA1, HSFA1-HSFA5, HSFA2-HSFA2, HSFA3-HSFA3, HSFA4-HSFA4, HSFA6-HSFA2, HSFA6-HSFA6, HSFA7-HSFA7, HSFA8-HSFA8, HSFB1-HSFB1, HSFB2-HSFB2, HSFB4-HSFB4, HSFC1-HSFC1, HSFC2-HSFC2	
Eudicots-Eudicots	Arabidopsis thaliana-Solanum lycopersicum	HSFA1-HSFA1, HSFA2-HSFA2, HSFA3-HSFA3, HSFA4-HSFA4, HSFA5-HSFA5, HSFA6-HSFA6, HSFA6-HSFA7, HSFB1-HSFB1, HSFB2-HSFB2, HSFB3-HSFB3, HSFC1-HSFC1	

Our results indicate that gene duplication in HSF genes has been a frequent event during the evolution of plants, significantly contributing to their expansion and functional diversification (Fig. 2). Our results also suggest that HSFA4 and HSFA5 have a close genetic relationship, the origin of which may be related to the ancient duplication of HSFA1. It is possible that HSFA6 and HSFA7 originated from gene duplication, most probably derived from HSFA2. HSFA9 was proven to be derived from HSFA2 after the divergence of ancestral angiosperms. HSFB1 is considered to be the most ancient among the HSFB genes, and we predict that HSFB2 and HSFB4 derived from HSFB1 considering the close relationship between them.

Figure 2 (A) Synteny analysis between the subfamilies HSFA2, HSFA6, HSFA7, HSFA9 of seven representative plant species (Amborella trichopoda, Liriodendron chinense, Arabidopsis thaliana, Solanum lycopersicum, Oryza sativa, Sorghum bicolor, Zea mays). (B) Synteny analysis between the subfamilies HSFB1, HSFB2, HSFB4, HSFB5 of seven representative plant species (Selaginella moellendorffii, Amborella trichopoda, Liriodendron chinense, Arabidopsis thaliana, Solanum lycopersicum, Oryza sativa, Zea mays). (C) Synteny analysis between the subfamilies HSFA1, HSFA4, HSFA5 of eight representative plant species (Ginkgo biloba, Amborella trichopoda, Liriodendron chinense, Arabidopsis thaliana, Solanum lycopersicum, Oryza sativa, Sorghum bicolor, Zea mays).

Black, and blue lines indicate orthologous, and paralogous gene pairs respectively. The different colored circle represent HSF genes from different subfamilies. The name of the genes is inside the circle.

Divergence time analysis

The estimated divergence dates of HSFA2 and HSFA9 in eudicots are indicated in Fig. 3. The divergence time of those two gene subfamilies in this study ranges from 131 Mya to 155.2 Mya, which is within the period of Late Jurassic to Lower Cretacous. The estimated split time of the HSFC2 clade and HSFC1 in monocots is indicated in Fig. 4, and ranges from 125 Mya to 190.4 Mya, which is within the Jurassic and Lower Cretaceous periods. The time of the occurence of these gene duplications are consistent with the origin of the most recent common acestors of all living angiosperm, which likely be around 140–250 Mya (Magallón et al., 2015; Foster et al., 2017). Although uncertainty remains for other characters, our reconstruction of the differentiation time scale between gene subfamilies allows us to propose a new plausible scenario for the early diversification of angiosperms at genomic level. The origin and rapid diversification of angiosperms represent one of the most intriguing topics in evolutionary biology (Sauquet & Magallón, 2018), and the evolution research of this gene family (such as the origin, expansion and loss of genes) provides an unprecedented opportunity to explore remarkable long-standing questions that may hold important clues toward understanding present-day biodiversity and adaption to different environments.

Figure 3 A dated phylogenetic reconstruction for the subfamilies HSFA2 and HSFA9.

Red ovals indicate gene duplication events. The divergence time of HSFA2 and HSFA9 are marked with red. The blue numbers on each node refer to the mean time to MRCA estimates; the blue numbers in parentheses on each node refer to the 95% highest posterior density intervals.

Figure 4 A dated phylogenetic reconstruction were done for the subfamilies HSFC1 and HSFc2. Red ovals indicate gene duplication events.

The divergence time of HSFC1 and HSFc2 are marked with red. The blue numbers on each node refer to the mean time to MRCA estimates; the blue numbers in parentheses on each node refer to the 95% highest posterior density intervals.

Discussion

Previous phylogenetic studies of the HSF gene family in plants have provided valuable insights into its evolutionary history (Scharf et al., 2012; Wang et al., 2018). However, the limited sampling of pteridophytes, gymnosperms and basal angiosperms have left unresolved questions regarding the origin of subclasses in the HSF gene family and their phylogenetic relationship and gene expansion patterns in different taxa. HSFs play a key role in the adaptation of plants to changing habitats and environmental stressors. Our understanding of land plant evolution at a genetic level in relation to environmental changes has also been hindered by sampling limitations (Rensing et al., 2008; Banks et al., 2011; Scharf et al., 2012; Nystedt et al., 2013; Lin et al., 2014; Wang et al., 2018; Lohani et al., 2019). Although ongoing plant genome projects will certainly uncover additional species or family-specific deletions and duplications, the general features are likely not to change (Thalmann et al., 2019). In this study, the diversity and number of plants examined allowed us to examine the evolutionary history of this gene family in a broader taxonomic context. Our phylogenetic analyses revealed a divergence of HSF subfamilies and independent evolution in plants, especially in angiosperms. It is still a big challege for multi-alignment of genomes to recognize the potential syntenic relationships due to the ubiquity of ancient and recent polyploidy events, as well as smaller scale events that derive from tandem and transposition duplications (Lynch & Conery, 2000; Bowers et al., 2003; Tang et al., 2008; Schranz, Mohammadin & Edger, 2012). However, thanks to a combination of phylogenic analyses and synteny analysis in this study, our results have scratched the surface of just how gene expansion in different land plant taxa occurred. Our results show that recent duplication events are mostly contributed to the puzzle clades (HSFA2, HSFA6, HSFA7, HSFA4, HSFA5) with members from other groups snuck in.

Our studies on different members of the HSF gene family from pteridophytes and gymnosperms reveal that this gene family is quite complex in terms of gene numbers and sequence diversity. We identified four subfamilies of HSFs (HSFA1, HSFA2, HSFB1, HSFB4) across candidate HSFs in six species of pteridophyte, and five subfamilies of HSFs (HSFA1, HSFA2, HSFB1, HSFB2, HSFB4) from 16 species of gymnosperm. Though the number of HSFs in pteridophytes and gymnosperms is significantly less than in angiosperms, the number of HSFA1 and HSFB1 genes in those taxa was higher than in angiosperms. It is assumed that pteridophytes and gymnosperms preferred to reserve more ancient members in HSFs subfamily. The HSFA1 and HSFB1 subfamilies in pteridophytes and gymnosperms separately formed more than one clade on a phylogenic tree with low support without clustering together, consistent with the findings that more ancient duplication events affect more distant taxonomic comparisons (Bowers et al., 2003). Only two genes (SelmoHSFB1b and SelmoHSFB4) in S. moellendorffii appeared to be the result of duplication events detected in a syntenic analysis. These findings indicate that HSFA1, HSFA2, HSFB1 and HSFB4, which were already commenced in the ancestor of all land plants, are ancestral gene groups.

Gymnosperm lineages were considerably diverged during the Late Carboniferous to the Late Triassic periods, and were dominant through most of the Mesozoic period (Bowe, Coat & DePamphilis, 2000; Chaw et al., 2000). However, massive extinction occurred in the Cenozoic period caused gymnosperm genera to diversify slower than angiosperms (Crisp & Cook, 2011). Ancient gene subfamilies, such as HSFB1 and HSFA1, experienced differentiation and variation over a long period of time, which may explain the molecular phylogenetic uncertainty within gymnosperms. Ancient WGDs have been probably inferred in the ancestry of all extant seed plants, and angiosperm and gymnosperm lineages have experienced additional rounds of WGD (Cui et al., 2006; Barker et al., 2008; Soltis, Visger & Soltis, 2014; Jiao et al., 2014; Li et al., 2015; Cannon et al., 2015). Although no syntenic gene was detected in gymnosperms, two or more genes from different subclasses form strongly supported clades (such as PintaHSFA1a and PintaHSFA2, AbifiRHSFB1a and AbifiRsfB4a), so the absence of syntenic genes in gymnosperms may be a result of the incomplete data sampling, or relatively lower quality of currently available assembly in gymnosperms. Alternatively, ancient interspersed segmental duplication of those genes occurred recently could be detected though phylogenetic and synteny analyses.

In angiosperms, the HSF gene family has undergone extensive duplications that have given rise to complicated orthology, paralogy, and functional heterology relationships. Our results showed that the diversity and number of HSF genes in angiosperms is remarkably higher than in other much earlier diverged taxa. We also observed a higher diversity and number of multiple paralogous and ortholog genes in angiosperms. Most of the gene copies generated by WGD events have been lost due to fractionation and subsequent “postdiploidization” or malfunctionalization (Jiao et al., 2011). Gene duplication is an important mechanism for genomic innovation (Li et al., 2016), and the functional divergence of duplicate genes retained from whole genome duplication (WGD) is thought to promote evolutionary diversification. Recent WGDs occurring in angiosperms, especially lineage-specific WGDs, have allowed the expansion and variation of HSFs, which supported by previous studies in Fagopyrum tataricum (Liu et al., 2019) and genus Brassica (Lohani et al., 2019). The results of a synteny analysis confirmed that the HSFA9 subfamily was only present in eudicots which derived from HSFA2, and HSFC2 genes were only present in monocots which derived from HSFC1. New genes originated from the divergence of paralogue genes, which resulted from duplication events. These two duplication events occurred early in angiosperm divergence, consistent with angiosperm radiations occurring in the Late Jurassic and Lower Cretaceous periods (Li et al., 2019). Approximately 132 Mya ago, angiosperms underwent rapid radiation to become the most diverse and successful plant group on land (Sanderson & Donoghue, 1994). The co-occurrence of retained duplication events with key processes in biological innovations underlines the importance of this crucial mechanism (Airoldi & Davies, 2012). The HSFB3 and HSFB5 subfamilies were found to be absent in monocots, but present in most basal angiosperms and eudicots. We hypothesize that HSFB3 and HSFB5 were thoroughly lost in the most recent common acestors of monocots, yet, their origin and evolutionary history remain poorly understood. We speculate that those gene loss events occurred from divergence early in angiosperm history. The above results indicate that species not only experienced rapid early radiation, diversification and mass extinction (Deenen et al., 2010; Meredith et al., 2011; Wickett et al., 2014; Zeng et al., 2014; Li et al., 2019), but also that genes went through expansion, diversification, and loss. After the divergence of angiosperms, eudicots and monocots experienced independent evolutionary processes.

Conclusions

The progressive increased data of whole geome assembly from different phylogenetic lineages has advanced our evolutionary understanding of gene families. Our comprehensive analysis reveals that the diversification of HSFs in plants resulted from extensive gene duplications and gene loss during the evolution and diversification of land plants. Lineage-specific expansions in angiosperms, especially in eudicots and monocots, may reflect the potential evolutionary advantage of flexibility in complex environments. The patterns of gene duplication and the evolutionary history of HSFs in plants provide novel insights into their diversity which facilitates the plant diversification, adaptation and evolution in various habitats. Our analyses provide essential insights for studying the evolutionary history of multigene families.

Supplemental Information

File S1 Accession numbers of HSF sequences used for the phylogenetic analysis

Click here for additional data file.

File S2 The source of genes used in this study

Click here for additional data file.

Figure S1 A detailed Maximum-Likelihood tree of HSFs constructed with the program RaxML

The classifications are based on HSFs annotations identified using the HEATSTER site (https://applbio.biologie.uni-frankfurt.de/HSF/heatster/).

Click here for additional data file.

Figure S2 A ML tree constructed by FastTree

The local support values were showed on tree.

Click here for additional data file.

Figure S3 A NJ tree tree was constructed by TreeBeST

The bootstrap values were showed on tree.

Click here for additional data file.

Figure S4 An NJ tree showing the phylogeny and classification of 670 HSFs sequences

HSFA, HSFB and HSFC are clustered into three main clades. The clade of subfamilies HSFA2/A6/A7, HSFA 3, HSFA 4, HSFA 5, HSFB2, HSFB3, HSFB5, HSFC1 and HSFC2, were shown over relevant branches with different colors. The three groups HSFA, HSFB, and HSFC were highlighted with shades of different colors.

Click here for additional data file.

Figure S5 A Maximum-Likelihood phylogenetic reconstruction for the group HSFA and HSFC

The branch and the genes’ name of distinct clusters of HSFA3, HSFA4, HSFA5, HSFA8, and HSFA9 were color coded. In the others, only the gene names were colored. The relevant support values are shown. The scale bar represents amino acid substitutions per site.

Click here for additional data file.

Figure S6 A Maximum-Likelihood phylogenetic reconstruction for the HSFB group

The branch and the genes’ name of distinct clusters of HSFB2, HSFB3, HSFB5 were color coded. In the others, only the gene names were colored. The relevant support values are shown. The scale bar represents amino acid substitutions per site.

Click here for additional data file.

Table S1 The species list used for HSF identification and analysis in this study

Click here for additional data file.

Table S2 The species list of HSFs data downloaded from Heatster

Click here for additional data file.

Table S3 Plant genomes used in this synteny analysis

Click here for additional data file.

Table S4 The genes used for phylogenetic tree construction

Click here for additional data file.

Table S5 HSF Gene pairs identified on syntenic blocks within different species

Click here for additional data file.

Table S6 HSF Gene pairs identified on syntenic blocks within different species

Click here for additional data file.

Table S7 Detailed information for the inferred duplication events within different species

Click here for additional data file.

Table S8 HSF Gene pairs identified on syntenic blocks between different species

Click here for additional data file.

Table S9 Detailed list of ortologous gene clusters between different species

Click here for additional data file.

Table S10 The source of genes used in this study

Click here for additional data file.

Additional Information and Declarations

Competing Interests

Author Contributions

Data Availability

The authors declare there are no competing interests. Dan Yang and Linping Meng all have been analyzed data. Because of her transfer, Dan Yang suggested the authorship change to Linping Meng.

Yiying Liao conceived and designed the experiments, performed the experiments, analyzed the data, prepared figures and/or tables, authored or reviewed drafts of the article, and approved the final draft.

Zhiming Liu performed the experiments, analyzed the data, prepared figures and/or tables, and approved the final draft.

Andrew W. Gichira analyzed the data, authored or reviewed drafts of the article, and approved the final draft.

Min Yang analyzed the data, prepared figures and/or tables, and approved the final draft.

Ruth Wambui Mbichi analyzed the data, authored or reviewed drafts of the article, and approved the final draft.

Linping Meng analyzed the data, prepared figures and/or tables, and approved the final draft.

Tao Wan conceived and designed the experiments, authored or reviewed drafts of the article, and approved the final draft.

The following information was supplied regarding data availability:

The raw data is available in the Supplemental File.

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
