# Peer review of "Deep evaluation of the evolutionary history of the Heat Shock Factor (HSF) gene family and its expansion pattern in seed plants"

_PeerJ, doi:10.7717/peerj.13603_

## Round 0.1 · original submission · Major Revisions

Despite the numerous and substantial requests of the reviewers I want to encourage the authors to follow all the requests and to improvise and resubmit the article with all the necessary modifications.

Reviewer 1 ·

Basic reporting

no comment

Experimental design

no comment

Validity of the findings

no comment

Additional comments

This manuscript identified HSFs in some species which were sequenced more recently, and investigated the evolutionary relationship of some HSF subfamilies. The HSF subfamily classification should also be induced from the phylogenetic tree, with a comparison to the annotation from HEATSTER website. Thereafter, some results should be thoroughly investigated and verified according to new classification results.

Major issues:
1. The scientific question should be more clearly and detailed (lines 80-81). Contrary conclusions, or insufficient evidences reported by previous studies should be indicated.

2. The author should precisely show how the HSF was identified using transcriptome data, with or without RNA-seq assembly?

3. The subfamily classification in table 1 is derived from the HEATSTER. The classification should be induced from the phylogenetic tree, instead of the HEATSTER. This is may be the reason that your result that HSFs from Chlorophyta were divided into HSFA (I didn't find these HSFs in figure 1). The previous study reported the HSFs from Chlorophyta were more closely related with HSFB in plants and HSFs in animal.

4. The phylogenetic tree in figure 1 has no bootstrap values supporting the robust of the tree. Other phylogenetic trees constructed by different model or methods should be included to support your classification and evolutionary results. So, the first result “The phylogeny and evolution of HSFs in land plants” should be verified and reconstructed.

5. In result parts of “Divergence time analysis”, why were only HSFA2 and HSFA9, and HSFC2 and HSFC1 included? How about the other subfamilies?

Some Minor issues:
59 line, “Thermo tolerance” should be “Thermotolerance”

60 line, “They (HSF) functionalized as molecular chaperones in protein folding and assembly to protect cells”. This is the HSP’s functions, not the HSF.

68 line, “transcription active region” of the HSF should also be included.

160 line, a total of 46 species in Table S1 and S2, instead of 44, were used for HSF identification. In addition, why were these species divided into two supplemental tables? The species which were not used in previous study should be indicated.

185 line “The deep divergence of HSFA1 in pteridophyta and gymnospermae, indicated that HSFA1 diversified before the radiation of seed plants” showed be exhibited as a figure. I think the separated clades of HSFA1 should be classified as other subfamilies (not evolutionary divergence) based on a robust phylogenetic tree, instead of HEATSTER. See the major issue 3.

254 line “The results indicate the ancient HSF gene duplications were not easy to be detected, because most duplicates have been lost”. The author should give the evidences or references supporting the “most duplicates have been lost”

In table 2, the sequence identities of the duplicated HSF should be exhibited to help readers for the understanding of duplication.

In table 3, the definition and the identification methods of the orthologous were not clearly discribed.

402 line, no evidences were provided to support the “gene loss and sub- or neo-functionalization during the evolution and diversification of land plants”

The resolution of the figures 2 and 3 should be improved.

Reviewer 2 ·

Basic reporting

The basic reporting of the article is ambiguous with several unclear statements, which often confuses the actual findings. The word 'deep' in the title itself is ambiguous. Moreover, the expansion of HSFs was inferred in 'seed plants' as appears from the title, but the sampling of seed plants analyzed by the authors may not be sufficient to make a blanket statement. In the article, authors have often mixed Results with Discussions, though a separate Discussion section existed. Some crucial literature references were missed out by the authors, which might have changed their perception of the evolution of HSFs in seed plants. There are also problems with the representation of Tables. The contents were not very clearly understood as far as the paralogs and orthologs are concerned. As far as the writing is concerned, the authors were very casual in checking the typos and English usages in the sentences throughout the manuscript. That results in so much annoyance that I could barely read beyond the Results section. I am sorry, though it sounds a bit harsh, I could not stop making this comment for a scientific review for a reputed journal like PeerJ. This critical assessment might enrich authors to improvise on the sections where they need to strengthen.

Experimental design

The experimental design though is okay, but somewhere falls short. Broadly, authors have compared the HSF proteins from a few groups of plant taxons, which could have expanded further with a large number of sequenced genomes available. Though it has been often mentioned by the authors as a comprehensive account of HSF, it is merely more than 600 HSF sequences, when there are 2.4 K sequences from 103 species already available in the Pfam database. The research question on the evolution of the HSF genes was also not well defined. Why do authors need to reanalyze the evolution of HSF genes may clearly be stated. Additional methods and approaches may need to be supplemented in order to really provide valuable and novel information on HSFs evolution and functionalities in plants.

Validity of the findings

The findings are not very novel and aren't explicitly validated with robust experimental results. The theme of research and conclusion looks okay, but without sufficient data to support it, the conclusions perhaps appear weak and overstated.

Additional comments

The manuscript may be checked thoroughly. If required, it can be checked through professional editors to submit a manuscript with minimal flaws. The image qualities and information depicted can be improved further.

Annotated reviews are not available for download in order to protect the identity of reviewers who chose to remain anonymous.

---

## Round 0.2 · Minor Revisions

The authors have addressed the concerns of reviewers but the manuscript needs additional editing (mostly for English) to be suitable for publication.

---

## Round 0.3 · accepted · Accept

According to the reviewer and my opinion, the review submitted by the authors is now significantly improved and suitable for publication in the journal. the job is then accepted.

Reviewer 1 ·

Basic reporting

No comment

Experimental design

No comment

Validity of the findings

No comment